# Adherence to Guideline-Directed Medical Therapy in Hospitalized Older People with Heart Failure at Discharge and 3-Month Follow-Up

**DOI:** 10.3390/jcm14196928

**Published:** 2025-09-30

**Authors:** Renee C. M. A. Raijmann, Melanie Haverkamp, Manon G. van der Meer, Wilma Knol, Cheyenne C. S. Tseng, Carolina J. P. W. Keijsers, Marielle H. Emmelot-Vonk, Huiberdina L. Koek

**Affiliations:** 1Department of Geriatrics, UMC Utrecht, Utrecht University, 3584 CX Utrecht, The Netherlandsh.l.koek@umcutrecht.nl (H.L.K.); 2Department of Geriatrics, Jeroen Bosch Ziekenhuis, 5223 GZ ‘s-Hertogenbosch, The Netherlands; 3Department of Cardiology, UMC Utrecht, 3584 CX Utrecht, The Netherlands

**Keywords:** guideline directed medical therapy, heart failure, aged, optimal drug dosage

## Abstract

**Objectives:** Heart failure is one of the leading causes of hospitalization in older adults. Guideline-directed medical therapy (GDMT) reduces the risk of decompensation and hospitalization, though it is challenging to implement GDMT in this group. Therefore, this study evaluated adherence to the 2021 ESC heart failure guideline in older patients and explored reasons for guideline deviations. **Methods:** A retrospective cohort study was performed in older patients (70+ years) with decompensated heart failure (ejection fraction < 50%) admitted to the cardiology or geriatrics department at a tertiary hospital (May 2022–September 2023). Data on GDMT-drug use, dosage, and reasons for guideline deviations were collected at discharge and three months post-discharge. Additionally, associations between GDMT non-adherence and factors such as age, frailty, comorbidities, and admission specialty were analyzed. **Results:** A total of 60 patients were included (mean age 79 years, 40% women, median EF 34%). The four GDMT drugs were prescribed to 15% of patients at discharge and 26% at follow-up, and 3% of the patients received the target dose for all drugs. Older patients (>80 years) received GDMT less frequently at discharge compared to younger patients (4% vs. 26%, *p* = 0.03). Though this difference was resolved at follow-up. The other studied factors were not significantly associated with GDMT adherence. Common reasons for guideline deviations were adverse effects, contraindications, reduced life expectancy, and postponed treatment. **Conclusions:** Adherence to GDMT in older heart failure patients is low due to several reasons, such as relevant contraindications. Physicians should carefully balance the risks and benefits of the guideline versus individual benefit, considering life expectancy and individual care goals.

## 1. Introduction

The number of older patients with heart failure is expected to increase significantly in the near future [1,2]. One of the challenges that physicians face in managing this growing patient group is how to implement guideline-directed medical therapy (GDMT). The recently updated guideline, provided by the European Society of Cardiology (ESC), recommends treating all patients with heart failure with a reduced ejection fraction (HFrEF) with four drug groups: beta-blockers (BBs), mineralocorticoid antagonists (MRAs), angiotensin receptor neprilysin inhibitors (ARNIs) or angiotensin-converting enzyme inhibitors (ACE-Is), and sodium-glucose cotransporter-2 inhibitors (SGLT2-Is), as depicted in Figure 1 [3].

Using all four drugs simultaneously has been associated with an improved life expectancy and a reduction in heart failure-related hospital admissions in previous studies [4]. However, implementing quadruple therapy in clinical practice is challenging, especially in older or frail adults [5,6,7,8]. Recent research indicated that there are significantly lower prescription rates of GDMT medications in older patients compared to younger patients [9,10,11]. This undertreatment seems to be associated with multimorbidity, frailty, and lower kidney function [12,13]. Both the beneficial effects and safety of GDMT have not been well established in frail and old patient populations because they are often not or underrepresented in clinical trials [6]. This limited representativeness might be one of the reasons why physicians deviate more often from the guideline for this group.

Furthermore, implementation of GDMT often introduces polypharmacy (use of >5 drugs) or hyper-polypharmacy (use of >10 drugs). Polypharmacy is associated with under-prescribing, prescribing cascades, drug-related hospital admissions, and higher mortality rates [12]. While arguments can be made that polypharmacy in heart failure patients is inevitable and can be appropriate, appropriateness should be evaluated in a multidisciplinary setting where comorbidities, individual side effects, quality of life, and individual treatment goals are also considered [12].

Yet, it might be argued that especially older and frail patients may have the most to gain from intensive treatment (also known as the risk-treatment paradox) and that withholding GDMT from these patients due to concerns for adverse effects is a form of clinical inertia [13,14]. Previous research also demonstrated inter-physicians’ variance, namely significantly different GDMT-drug prescription rates between cardiologists in similar patient populations [15]. These findings suggest that there is practice variety [14,15]. It can be assumed that overcaution may also contribute to lower prescription rates in older and frail patients.

Literature on guideline adherence is lacking for (frail) older heart failure patients. Based on previous research, we hypothesized that guideline adherence in this population is low and could be caused by various reasons. Therefore, this study aimed to investigate adherence to the ESC heart failure guideline (2021) in older patients with heart failure with an ejection fraction under 50%. Additionally, reasons for guideline deviations were evaluated, and measurable determinants (such as age, frailty, comorbidity, and admission specialty) that may be associated with guideline adherence were investigated.

## 2. Materials and Methods

### 2.1. Study Design

This retrospective cohort study in older patients admitted to the hospital with heart failure studied adherence to GDMT. In accordance with the 2021 ESC guideline, GDMT means treatment with four drugs: BBs, MRAs, ARNIs or ACE-Is, and SGLT2-Is (Figure 1) [3]. The prescription rates were evaluated at discharge and three months post-discharge at the University Medical Center Utrecht in the Netherlands in the period between May 2022 and 15 September 2023.

### 2.2. Study Population

Eligible patients were individuals aged 70 years and older who were admitted to the cardiac or geriatric department with decompensated heart failure. Other inclusion criteria were an ejection fraction < 50% on echocardiography and a hospital admission that lasted at least 24 h. Patients with a mildly reduced ejection fraction (HFmrEF, ejection fraction of 41–49%) were also included in this study, similar to previous studies [5,15], as this group also contains patients with HFrEF prior to treatment with heart failure drugs. Though it is important to note that patients with HFmrEF and HFrEF are treated differently according to the ESC guideline. For the HFmrEF population, only the SGLT2-I is recommended with a Class I recommendation [3]. For the other three drugs (BB, MRAs, and ARNIs or ACE-I), the recommendations are less strong (Class IIb recommendation) [3].

To improve comparability between patients admitted to the cardiac and geriatric departments, patients requiring continuous hemodynamic monitoring or those with a left ventricular assist device were excluded. Patients who died during admission were also excluded.

### 2.3. Exposure

At the beginning of the study (May 2022), patients were admitted to a geriatric or cardiac ward based on their medical history. Patients admitted to either the cardiology or geriatrics department received usual care as provided by their respective care teams. Consultation between physicians was only performed upon request.

In July 2023, a multidisciplinary program was initiated, which involved admitting all heart failure patients aged 70 years and older to the geriatric department after assessment by a cardiac and geriatric physician in the emergency room. After admission, patients received usual care provided in a geriatrics department in combination with cardiac co-management. This meant that the cardiology and geriatrics physicians collectively evaluated pharmacological therapy, the need for any additional investigations, and long-term care plans. These long-term care plans included advance care planning and outpatient clinical follow-up. If, at the time of admission, there was no space in the geriatrics department, or joint assessment in the ER was not feasible, patients were admitted to the cardiac department with involvement of a geriatric consultant team.

### 2.4. Ethics

The local institutional review board gave approval for a waiver to obtain informed consent (reference number 23U-0319) given the anonymity of data collection and the non-interventional nature of the study.

### 2.5. Data Collection

Data were retrospectively extracted from patients’ electronic medical records. The collected baseline characteristics included general demographics, the estimated glomerular filtration rate, ejection fraction, medical history, heart failure etiology, number of medications, cardiovascular medications, a clinical frailty score (CFS), fall risk, and residential status.

The researchers additionally collected data on the use and dosage of the four GDMT drugs. In case quadruple therapy was not prescribed, the reasons for the guideline deviation that the physician had formulated in the patient’s records were classified in categories: (1) postponed treatment (meaning the physician thinks it is too early to start a GDMT drug but means to initiate the drug in the future); (2) adverse effects; (3) contraindications; (4) reduced life expectancy; (5) other reason(s); (6) no reason reported.

Drug doses were calculated as a percentage of the recommended target dose specific to the drug. Target dosages were derived from the ESC heart failure guideline and are listed in Appendix A. A drug dosage of at least 50% of the target dose was considered optimal medical therapy [16].

The researchers (MH and RR) retrospectively estimated the CFS based on information regarding mobility and care dependence documented by nursing staff or physicians. In case MH and RR rated differently, the discrepancy in CFS was resolved through discussion, which led to consensus on the appropriate CFS. This method of retrospectively assessing frailty using the CFS has been validated in previous studies [17].

To evaluate polypharmacy, all medications prescribed at discharge were counted using the Anatomical Therapeutic Chemical (ATC) codes [18]. Multiple medications with the same ATC code were counted as one drug. For example, furosemide and spironolactone fell into the ATC code diuretics and were counted as one drug. Medication taken as needed, dermatologic ointments, and artificial tears were not counted.

The Charlson comorbidity index [19] was also calculated retrospectively based on the medical history documented in the patients’ records. This is a prognostic index based on a patient’s comorbidities, including, among others, history of myocardial infarction, heart failure, cerebrovascular incidents, diabetes, kidney disease, and malignancies.

The last assessment tool researchers collected was the Johns Hopkins in-hospital fall risk assessment tool [20]. This data could be collected directly from the patient records, as it was common practice for the nurses to document this score on admission. This score is based on age, history of falls, (in)continence, medication use, mobility impairments, cognitive impairments, and need for equipment that tethers the patients, such as lines, catheters, or drains.

### 2.6. Statistical Analysis

Baseline characteristics were presented using descriptive statistics. Baseline differences between the two admission specialties were examined through the Pearson Chi-square test, Fisher’s exact test, and the Mann–Whitney U test, where appropriate.

Differences in prescription rates between time points (discharge vs. 3-month follow-up) were calculated using McNemar’s test. Furthermore, we also analyzed whether GDMT prescription rates differed between HFrEF and HFmrEF populations. For these analyses, a Pearson Chi-square test or Fisher’s exact test was used where appropriate.

The association between GDMT adherence and patient- and care-related factors was studied. These factors were defined as age (≥80 vs. <80 years), sex (male vs. female), CFS (≥5 frail vs. <5 non- or mildly frail) [17], Charlson Comorbidity index (≥4 vs. <4) [19], John Hopkins in-hospital fall risks assessment tool (high vs. low–medium fall risk) [20], number of drugs used at discharge (≥10 hyper polypharmacy vs. <10 no polypharmacy and minor polypharmacy) [21], and admission specialty (cardiology vs. geriatrics). These were analyzed using Fisher’s exact test due to low counts in cells. All analyses were performed with the Statistical Package for Social Sciences (SPSS) version 29.0 [22]. A two-sided *p*-value of 0.05 was considered statistically significant.

## 3. Results

### 3.1. Patient Characteristics

A total of 60 patients were included in this study. Patients’ selection and follow-up are illustrated in Figure 2. In total, 82% of patients who were alive at the end of the study had completed follow-up. The loss of follow-up was caused by patients who were treated in outpatient clinics of other nearby hospitals. At the three-month follow-up, 23% of patients had died.

Baseline characteristics are presented in Table 1. The mean population age was 79.0 years (interquartile range 75.3 to 85.0), with 40% being women and 58% being frail. Patients admitted to the geriatric ward were significantly older (∆5.6, *p* < 0.01), had a higher Charlson comorbidity index (∆0.6, *p* = 0.04), had a higher fall risk (∆11%, *p* < 0.01), and were more often frail (∆12%, *p* = 0.02) compared to patients admitted to the cardiological ward.

### 3.2. Medication Use

#### 3.2.1. Discharge

As shown in Table 2, at discharge, 15% of patients were treated with quadruple therapy, and 8% of patients did not receive any of the four GDMT drugs. An MRA was most prescribed (77%), followed by a BB (60%), then an ACE-I (45%), and lastly an SGLT2-I (42%). When comparing the HFrEF and HFmrEF patients, we found that the SGLT2-I prescription rates were higher in the HFrEF group (53% vs. 23%). The prescription rates for the other drugs did not importantly differ between the HFrEF and HFmrEF groups.

Patients aged 80 years and older were less frequently treated with quadruple therapy than younger patients (3% versus 27%, *p* = 0.03); see Table 3. There was no statistically significant difference in quadruple therapy prescription rates when comparing the geriatric and cardiac departments (12% versus 18%, *p* = 0.72) and frail and non-frail to mildly frail patients (11% versus 20%, *p* = 0.47). For the other investigated determinants (sex, Charlson comorbidity index, polypharmacy, and fall risk), there were also no significant differences in quadruple therapy prescription rates (Appendix A).

#### 3.2.2. Three-Month Follow-Up

As shown in Table 2, at three-month follow-up, 26% of patients used all four drugs (15% at discharge), and there were no patients (0%) who did not receive any of the drugs (8% at discharge).

Patient-related factors (sex, age, frailty, Charlson comorbidity index, polypharmacy, fall risk, and admission specialty) that were studied showed no significant differences in GDMT use (Appendix A).

### 3.3. Optimal Medical Therapy: Target Doses

Figure 3 illustrates the proportion of patients who reached the recommended target dose for each drug type.

Optimal medical therapy (>50% of the target dose) was reached most often for an MRA (74%), followed by an ACE-I (71%). For BBs, 36% of patients received optimal medical therapy. SGLT2-Is are only available in one dose, which is the target dose. Therefore, the percentage of patients reaching optimal medical therapy was 100%. Only 3% of patients reached the optimal medical therapy dose for all four medications at discharge.

After three months, the percentage of patients receiving optimal medical therapy was similar to discharge for MRAs (70%) and BBs (35%). The percentage of patients receiving optimal medical therapy with an ACE-I appeared to be lower, but this was not significant (71% to 55%, *p* = 0.25). At follow-up, 8% of patients reached optimal medical therapy doses for all four medications.

### 3.4. Reasons for Guideline Deviations

The reasons for guideline deviations are shown in Figure 4. Adverse effects were the most frequent reason for discontinuing an ACE-I both at discharge and during follow-up, of which orthostatic hypotension was the most common (30%). Reported contraindications were documented allergies or hyperkalemia. Reported reasons for not prescribing BBs upon discharge included postponed treatment (21%) and reduced life expectancy (21%). After three months, the main reason was due to adverse effects (42%) such as orthostatic hypotension, bradycardia, and fatigue. Upon discharge, MRAs were not prescribed, mainly due to postponed treatment (29%) or reduced life expectancy (21%). For the 38 patients who completed follow-up, 16% of patients did not use an MRA. For half of these patients, the reasons for not using an MRA were not documented. For 33% of patients, MRAs were discontinued due to adverse events, mostly hyperkalemia. SGLT2-Is were often not prescribed at discharge due to postponed treatment (29%) and contraindications (26%), such as impaired renal function. At follow-up, the most frequent reason for not using an SGLT2-I was a contraindication (29%). Adverse events occurred less frequently in the case of SGLT2-I use (9% at discharge and 7% at follow-up). Reported adverse events were a urinary tract infection and orthostatic hypotension.

Overall, at three months follow-up, for 28% of all guideline deviations, the reason was not documented. Furthermore, the reason for 13% of guideline deviations was due to ‘postponed treatment’.

## 4. Discussion

In this study, we examined adherence to quadruple therapy with BBs, ACE-I/ARB, SGLT2-Is, and MRAs, also known as GDMT, and reasons for guideline deviations in an older patient population admitted for heart failure. We found that 15% of older (70+) patients used GDMT after a heart failure-related hospital admission. When divided by ejection fraction, 18% of the HFrEF group and 9% of the HFmrEF group received quadruple therapy. Optimal medical therapy dosing for all four medications was achieved in only 3% of patients. At three-month follow-up, 26% received quadruple therapy, and 8% of patients achieved optimal medical therapy dosages. Older age (80+) was associated with reduced adherence to GDMT. The most common reasons for guideline deviations were reasons such as adverse effects, postponed treatment, and contraindications, but frequently no reasons were found in the medical record. The adherence rates in this study were lower compared to the findings of the TITRATE-HF study [23]. They found that 44% of chronic and worsening HFrEF patients and 25% with HFmrEF were prescribed quadruple therapy at hospital admission. There are two explanations as to why prescription rates in the current study were lower [23]. Firstly, our patient population was older (78 vs. 71 years). Secondly, we did not exclude patients with a shorter life expectancy than one year. In contrast, similar results to our study were found in another recent observational study on quadruple therapy use in HFrEF patients aged 80 years and older (average age 82 years). In 10% of the patients, quadruple therapy was prescribed at baseline; after follow-up of 25 months, the prescription rate increased to 28% [24].

Another factor that might have negatively influenced the GDMT prescription rates in our study was the low prescription rate of GDMT in the included HFmrEF population. HFrEF patients received SGLT2-Is significantly more often than HFmrEF patients (53% versus 23%, *p* = 0.03), even though the guideline recommends the use of SGLT2-Is in both patient types. This might be caused by the fact that the recommendation for HFmrEF patients was published in an update of the ESC guideline in August 2023, which was during the inclusion period of our study (May 2022 to September 2023). Most patients finished their three-month follow-up period before the publication of this recommendation.

Regarding optimal medical therapy (dose > 50% of target dose), only 3% of patients in this study received all four drugs in this dosage. Previous research recognized this as challenging as well, with only 10 to 20% of patients reaching the target doses [9,15,25]. Though, in clinical practice, achieving a certain target dose is not a goal in and of itself. Variables such as blood pressure, heart rate, and laboratory findings are used to determine the optimal dose for each patient.

When evaluating GDMT adherence in the context of patient- and care-related factors, we found that older age was negatively associated with GDMT adherence, which is similar to previous studies [9,23,25,26]. Previous research also demonstrated that frail patients are less likely to receive GDMT compared to non-frail patients [7,27]. In our study, we did not find a statistically significant difference between frail and non-frail patients. Though, we observed that 11% of the frail patients received GDMT compared with 20% of the non-frail patients. Similarly, there was no significant difference in GDMT prescription rates between cardiology and geriatric specialists (18% vs. 12%), which contrasts with previous studies [9,28]. In both, the difference is likely not significant because of the small sample size.

This study found that reasons for guideline deviations were adverse effects and contraindications. Adverse effects and contraindications were also the main reasons for discontinuing GDMT drugs in other studies [13,25]. Another important reason to deviate from the guideline was a reduced life expectancy, which is also illustrated in the 23% three-month mortality rate in our patient population. In this patient population, especially, we need to consider a drug’s time to benefit and whether the benefit of the drug aligns with patient care goals, such as improved quality of life and functional independence.

Yet, we also found that some guideline deviations at the three-month follow-up were due to ‘postponed therapy’. It could be argued that these drugs should have been initiated more rapidly, as recent literature recommends rapid initiation and up-titration in the first 6 weeks of hospital discharge [4,29].

### 4.1. Strengths and Limitations

One of the strengths of the current study is that it is among the first studies to describe adherence to the quadruple therapy in an old population since the introduction of the updated ESC guideline. We analyzed prescription rates at discharge and three-month follow-up to account for further outpatient sequencing and up-titration of GDMT drugs. Furthermore, the study compared geriatricians and cardiologists as prescribers, which is relevant for gaining insight into the prescribing practices of different physicians. Lastly, this study provides a clear and comprehensive overview of reasons for guideline deviations.

There are also limitations to consider. Firstly, the sample size was very small, which led to insufficient power for detecting potentially important differences between the geriatric and cardiological wards, frailty, and other determinants. Secondly, during the three-month follow-up period, some patients had died, which further reduced the sample size and potentially caused attrition bias. However, our cohort is a natural cohort of patients with a short life expectancy. Additionally, our main aim was to describe GDMT adherence in daily practice, for which correction for confounding is not indicated. Lastly, we used a retrospective study design with data from previously documented information in medical records. This led to incomplete data and a restriction on which variables we could analyze, which could have resulted in information bias.

### 4.2. Recommendations

Although the guidelines are clear, it can be hard to follow the guidelines in older patients with heart failure. However, one should not hesitate to strive for this goal, given the risk-treatment paradox, meaning that the best results are also obtainable in the group most at risk of adverse outcomes. Initiating and up-titrating heart failure drugs should not be lengthily postponed due to concerns for adverse effects. However, it should be considered whether the added benefit of the drugs aligns with the patients’ care goals. As these patients have a limited life expectancy, the main care goals should be to improve quality of life, limit hospital admissions, and preserve functional independence.

Therefore, it is essential to conduct further research on the effectiveness of personalized GDMT in older and frail patients with heart failure, evaluating these outcomes. Especially the newer drug group SGLT-2 needs further investigation in larger groups of older patients to investigate the benefits and side effects. Some studies already suggest beneficial effects on cardiac events [30,31]. These future studies should systematically collect reasons for guideline deviations and could additionally perform interviews with participating physicians to gain a better understanding of the rationale.

## 5. Conclusions

In summary, this study showed low adherence to quadruple drug therapy in older heart failure patients, particularly in the oldest patients. The optimal medical dose was achieved even less frequently (3% and 8% at discharge and 3-month follow up, respectively). Reasons for guideline deviation were mainly well-considered. Using our findings, we urge physicians to weigh the potential harms and benefits of heart failure drugs in older patients, carefully considering their reduced life expectancy and individual care goals.

## Figures and Tables

**Figure 1 jcm-14-06928-f001:**
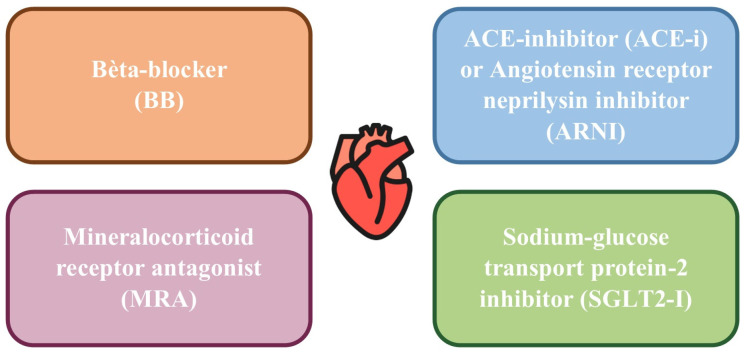
Guideline-directed medical therapy (GDMT) for heart failure with reduced ejection fraction according to the 2021 ESC guidelines. Abbreviations: SGLT2-I: sodium-glucose cotransporter-2 inhibitors.

**Figure 2 jcm-14-06928-f002:**
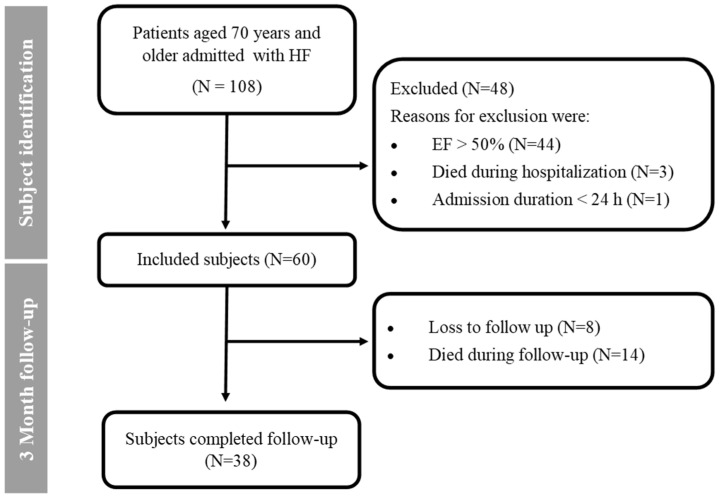
Flow diagram of subject selection and follow-up. Abbreviations: HF = heart failure, EF = ejection fraction, N = number of subjects.

**Figure 3 jcm-14-06928-f003:**
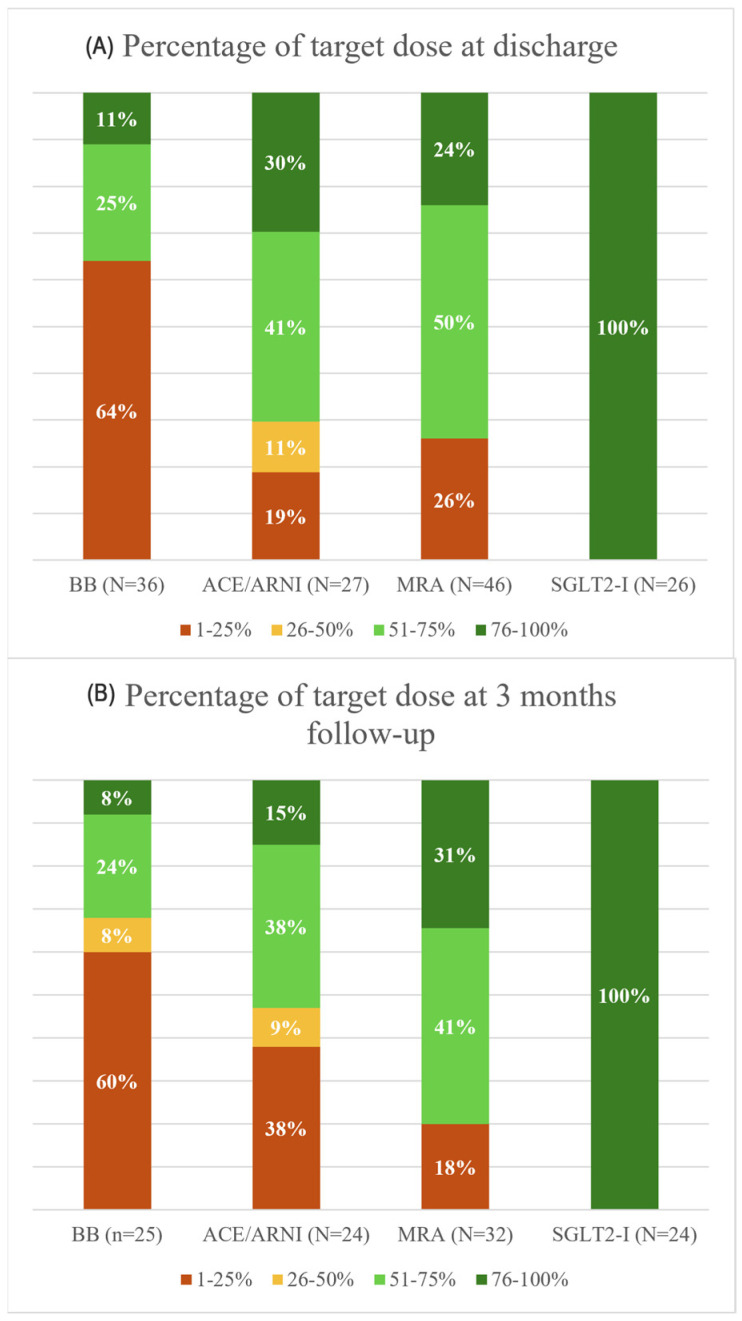
Target dose percentage for each GDMT drug at discharge and follow-up (FU). Figure 3 illustrates the percentage of patients who receive a specific percentage of the target dose (color group) for each GDMT drug at discharge (**A**) and after the three-month follow-up (**B**). The numbers depicted below each column in brackets are the absolute number of patients receiving the specific GDMT drug. Abbreviations: BB = beta-blocker, ACE-I/ARNI = ACE-inhibitor/angiotensin receptor neprilysin inhibitor, MRA = mineralocorticoid receptor antagonist, SGLT2-I = sodium glucose transport protein 2-inhibitor.

**Figure 4 jcm-14-06928-f004:**
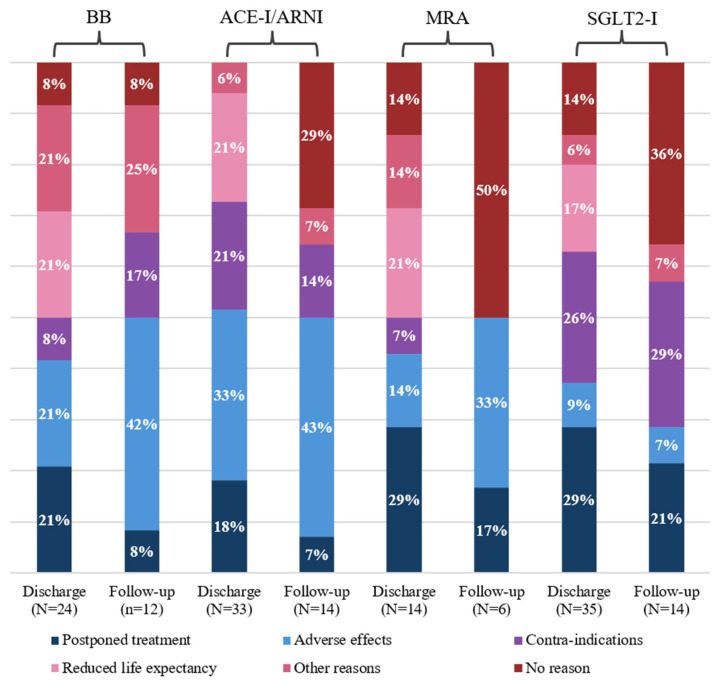
Reasons for GDMT non-adherence. Figure 4 illustrates the frequency of different reasons for guideline deviations (color groups) for each GDMT drug at discharge (baseline) and three-month follow-up. The numbers below the columns in brackets represent the absolute number of patients who did not receive the specific GDMT drug at discharge or three-month follow-up. Abbreviations: BB = beta-blocker, ACE-I/ARNI = ACE-inhibitor/angiotensin receptor neprilysin inhibitor, MRA = mineralocorticoid receptor antagonist, SGLT2-I = sodium glucose transport protein 2-inhibitor.

**Table 1 jcm-14-06928-t001:** Baseline characteristics of the study population.

	Total (N = 60)	Geriatric Ward (N = 32)	Cardiology Ward (N = 28)
**Demographics**			
Age, years	79.5 (75.3–85.0)	83.0 (78.0–87.0) *	77.4 (73.3–81.0) *
Female, %	25 (41.7)	16.0 (50.0)	9 (32.1)
Ejection fraction, %	33.8 (28.5–42.0)	36.0 (30.0–42.8)	31.6 (21.3–39.5)
Body mass index, kg/m^2^	25.4 (22.0–28.0)	24.9 (22.0–26.9)	25.9(22.4–28.8)
Charlson comorbidity index	6 (5–7)	6.4 (5.3–7) *	5.8 (4.3–7) *
**Medical history**			
Hypertension, %	31 (51.7)	18 (56.3)	13 (46.4)
Hypercholesterolemia, %	33 (55.0)	19 (59.4)	14 (50.0)
Myocardial infarction, %	30 (50.0)	16 (50.0)	14 (50.0)
Diabetes Mellitus, %	18 (30.0)	11 (34.4)	7 (25.0)
Chronic kidney disease (eGFR < 30)	6 (10.0)	3 (9.4)	3 (10.7)
Anemia, %	7 (11.7)	5 (15.6)	2 (7.1)
Atrial fibrillation or flutter, %	32 (53.3)	18 (56.3)	14 (50.0)
Heart valve disease, %	15 (25.0)	9 (28.1)	6 (21.4)
HFrEF, %	38 (63.3)	16 (50.0) *	22 (78.6) *
**Heart failure etiology**			
Ischemic, %	36 (60.0)	21 (65.6)	15 (53.6)
Valvular, %	5 (8.3)	2 (6.3)	3 (10.7)
Arrhythmia, %	4 (6.7)	1 (3.1)	3 (10.7)
**Polypharmacy**			
Minor polypharmacy (5–9), %	42 (70.0)	21 (65.6)	21 (75.0)
Hyper polypharmacy (≥10), %	14 (23.3)	8 (25.0)	6 (21.4)
**Cardiovascular drugs**			
Diuretics, %	50 (83.3)	26 (81.3)	24 (85.7)
Anti-arrhythmics, %	16 (26.7)	6 (18.8)	10 (35.7)
Antihypertensives, %	7 (11.7)	3 (6.4)	4 (14.3)
Anticoagulants, %	34 (56.7)	18 (56.3)	16 (57.1)
Thrombocyte aggregation inhibitors, %	21 (35.0)	10 (31.3)	11 (39.3)
**Clinical frailty scale** ^‡^			
Non-frail (1–2)	6 (10.2)	2 (6.3)	4 (14.8)
Mildly frail (3–4)	19 (32.2)	7 (21.9)	12 (44.4)
Frail (>5)	34 (57.6)	23 (71.9) *	11 (39.3) *
**Fall risk** ^‡^			
Low risk, %	10 (16.7)	2 (6.3) *	8 (28.6) *
Medium risk, %	32 (53.3)	16 (50.0)	16 (57.1)
High risk, %	17 (28.3)	14 (43.8) *	3 (10.7) *
**Residential status** ^‡^			
Home without care, %	45 (75.0)	23 (71.9)	22 (81.5)
Home with care, %	11 (18.3)	6 (18.8)	5 (18.5)
Residential home, %	3 (5.0)	3 (9.4)	0 (0)

Numbers are counts with percentages, unless otherwise mentioned. * *p* < 0.05, median and interquartile range. eGFR: estimated glomerular filtration rate. ^‡^ Total population n = 59; missing data for 1 participant.

**Table 2 jcm-14-06928-t002:** Adherence to GDMT at discharge and three-month follow-up.

	**Discharge**			**Three-Month Follow-Up**		
	**All (n = 60)**	**HFrEF (n = 38)**	**HFmrEF (n = 22)**	**All (n = 38)**	**HFrEF (n = 26)**	**HFmrEF (n = 12)**
**GDMT, per drug group**	**%**	**%**	**%**	**%**	**%**	**%**
ACE-I	45	45	46	63	65	58
ARNI	8	5	0	11	15	0
BB	60	22	14	68	69	67
MRA	77	29	17	84	89	75
SGLT2-I	42	20 *	5 *	63	73	41
**GDMT, number of drugs**						
All 4	15	18	9	26	31	17
3/4	30	32	27	34	38	25
2/4	27	24	32	32	27	42
1/4	20	16	27	8	4	17
0/4	8	11	1	-	-	-

Prescription rates for GDMT drugs and GDMT adherence stratified by ejection fraction. Statistical tests were performed to compare prescription rates between discharge and follow-up and between HFrEF and HFmrEF groups. * *p* < 0.05, abbreviations: HFrEF = heart failure with reduced ejection fraction, HFmrEF = heart failure with mildly reduced ejection fraction, BB = beta-blocker, ACE-I/ARNI = ACE-inhibitor/angiotensin receptor neprilysin inhibitor, MRA = mineralocorticoid receptor antagonist, SGLT2-I = sodium glucose transport protein 2-inhibitor.

**Table 3 jcm-14-06928-t003:** Prescription rates of all 4 GDMT drugs between patient characteristics.

	**Discharge**			**3-Month Follow-Up**		
**Characteristics**	**Number of Patients**	**Use of All 4 GDMT Drugs (%)**	** *p* ** **-Value**	**Number of Patients**	**Use of All 4 GDMT Drugs (%)**	** *p* ** **-Value**
Age						
70–79	30	27	-	22	32	-
80+	30	3	0.03 *	16	19	0.47
Sex						
Female	25	12	-	14	14	-
Male	29	17	0.72	24	33	0.27
CCI						
<4	3	33	-	3	67	-
≥4	57	14	0.39	35	23	0.16
History of HF						
Newly diagnosed HF	22	18	-	17	29	-
History of HF	38	13	0.71	31	24	0.73
John Hopkins fall assessment score						
Low–medium risk	42	17	-	27	22	-
High risk	17	12	1.00	10	30	0.68
Polypharmacy						
0–9 drugs	46	15	-	29	28	-
≥10 drugs	14	14	1.00	9	22	1.00
Clinical frailty score						
CSF < 5	25	20	-	22	23	-
CSF > 5	34	12	0.47	15	33	0.71
Admission specialty						
Cardiology	28	18	-	22	23	-
Geriatrics	32	13	0.72	16	31	0.71

Comparison of prescription rates of all 4 GDMT drugs for different patient characteristics. All comparisons were tested using a Fisher’s exact test. * *p* < 0.05, abbreviations: GDMT, guideline-directed medical therapy; CCI, Charlson Comorbidity index; HF, heart failure; CSF, clinical frailty score.

## Data Availability

The data presented in this study consist of anonymized patient information and are available from the corresponding author upon reasonable request. Public access is restricted due to ethical considerations and institutional data protection policies.

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
