# Peer review of "Adherence to Guideline-Directed Medical Therapy in Hospitalized Older People with Heart Failure at Discharge and 3-Month Follow-Up"

_jcm, 2025, doi:10.3390/jcm14196928_

Round 1
Reviewer 1 Report (New Reviewer)
Comments and Suggestions for Authors
Summary: The authors present an interesting and timely study that examines adherence to guideline-directed medical therapy in older patients with heart failure. This study retrospectively enrolled 60 patients with decompensated heart failure with an ejection fraction <50% (i.e. HFrEF & HFmrEF). GDMT adherence was collected at hospital discharge and 3-month follow-up. Only 15% and 26% of patients were prescribed GDMT at discharge and follow-up, respectively. The authors also provide data on the reasons for non-adherence to GDMT for each of the 4 major pharmacotherapies. Older patients were prescribed less GDMT than their younger counterparts in this cohort. The authors make note of the complexities of treating elderly patients with HF (polypharmacy, frailty etc,) and should be commended on the present study, which may further highlight this issue in the cardiology and geriatric medical communities.
General comments:
The authors have not provided any hypotheses in the methods section. The authors should either add hypotheses or comment on why they have been omitted from the manuscript.
Add p-values to Table 2 to make it clear which findings are significant and which are not.
Some of the presentation of the results is unclear, specifically, the numerical data not presented in the figures. It would be preferable if the data of older vs younger patients were presented in tabular form to make their findings clearer to the reader, given that this important finding was mentioned in the abstract but only in text in the results. Moreover, Table 2 states “Prescription rates for GDMT drugs and GDMT adherence stratified by ejection fraction.” Yet the results presented appear to be the total cohort and not stratified by ejection fraction. The authors should rectify this and include this data stratified by ejection fraction.
The authors state that only patients admitted with decompensated HF were included retrospectively in the study, yet the supplementary figure shows acute and chronic HF at discharge and at 3-month follow-up. Were these chronic HF patients clinically compensated at discharge and the acute HF not clinically compensated at discharge? The authors should comment on this for further clarity.
Do the authors have access to GDMT adherence on admission? This would be pertinent data to include if available, given GDMT reduces HF hospital admissions/decompensation.
Specific comments:
Figure 1 – SGLT-2 should include “inhibitors”
Line 100 there appears to be a citation issue
The authors interchange hyperpolypharmacy and major polypharmacy. Recommendation to use hyperpolypharmacy consistently throughout the manuscript.
Table 1 – Antiarrhythmica should be antiarrhythmics
Line 210/211 appears to be incomplete.
Author Response
Summary: The authors present an interesting and timely study that examines adherence to guideline-directed medical therapy in older patients with heart failure. This study retrospectively enrolled 60 patients with decompensated heart failure with an ejection fraction <50% (i.e. HFrEF & HFmrEF). GDMT adherence was collected at hospital discharge and 3-month follow-up. Only 15% and 26% of patients were prescribed GDMT at discharge and follow-up, respectively. The authors also provide data on the reasons for non-adherence to GDMT for each of the 4 major pharmacotherapies. Older patients were prescribed less GDMT than their younger counterparts in this cohort. The authors make note of the complexities of treating elderly patients with HF (polypharmacy, frailty etc,) and should be commended on the present study, which may further highlight this issue in the cardiology and geriatric medical communities.
Answer:
Dear reviewer, we would like to thank you for your positive evaluation of the paper and agree on the importance of the subject of the manuscripts topic. We value the time and effort you have put into reviewing our manuscript and feel that your comments have improved our manuscript.
General comments:
- The authors have not provided any hypotheses in the methods section. The authors should either add hypotheses or comment on why they have been omitted from the manuscript.
Thank you for making this suggestion, we agree that it is important to state an hypothesis to the manuscript. Therefore we have added this in the last paragraph of the introduction section.
- Add p-values to Table 2 to make it clear which findings are significant and which are not. Moreover, Table 2 states “Prescription rates for GDMT drugs and GDMT adherence stratified by ejection fraction.” Yet the results presented appear to be the total cohort and not stratified by ejection fraction. The authors should rectify this and include this data stratified by ejection fraction.
Answer:
We agree that the prescription rates stratified by ejection fraction are important data and have added them to the manuscript. Additionally, we performed statistical tests to determine whether there were any statistically significant differences between the prescription rates for discharge and follow up, and for HFrEF and HFmrEF patients. We also added a few sentences to the paragraph on statistical analyses in the methods section.
As only 1 significant difference was identified within table 2 we chose to highlight this in bold text and with an asterix, instead of providing the individual p-values.
- Some of the presentation of the results is unclear, specifically, the numerical data not presented in the figures. It would be preferable if the data of older vs younger patients were presented in tabular form to make their findings clearer to the reader, given that this important finding was mentioned in the abstract but only in text in the results.
We agree that this finding was important and would be more clear if presented in tabular form. In our old version of the manuscript, this data was presented in supplementary figureS1C. To enhance the comprehensibility of this analysis and other researched characteristics we have added table 3 to the main manuscript.
- The authors state that only patients admitted with decompensated HF were included retrospectively in the study, yet the supplementary figure shows acute and chronic HF at discharge and at 3-month follow-up. Were these chronic HF patients clinically compensated at discharge and the acute HF not clinically compensated at discharge? The authors should comment on this for further clarity.
Dear reviewer, we agree this wording is confusing. What we meant with the distinction between acute and chronic HF is whether a patient had a history of heart failure before admission or if they presented with acute heart failure without previous history of the disease. Therefore we have changed the wording in the supplementary figure from “chronic HF” and “acute HF” to “history of HF” and “newly diagnosed HF”.
- Do the authors have access to GDMT adherence on admission? This would be pertinent data to include if available, given GDMT reduces HF hospital admissions/decompensation.
Dear reviewer, thank you for this excellent suggestion, we have considered it extensively with our research team. We agree that GDMT (non)adherence on admission might have led to a possible hospital admission. However, we do not have access to this data currently. Furthermore, we fear that the GDMT prescription rates at admission may be even lower due to the large number of patients with no previous history of heart failure (22/60 patients). As these patients had no previous history of heart failure they probably received heart failure medication less frequently, leading to an underestimation of GDMT use on admission. Therefore, we feel that GDMT use at discharge is a more representative measurement of GDMT adherence in this population.
Specific comments:
- Figure 1 – SGLT-2 should include “inhibitors”
Answer: we have amended figure 1
- Line 100 there appears to be a citation issue
Answer: correct, we have corrected this issue.
- The authors interchange hyperpolypharmacy and major polypharmacy. Recommendation to use hyperpolypharmacy consistently throughout the manuscript.
Answer: we have changed major polypharmacy to hyperpolypharmacy in the entire manuscript.
- Table 1 – Antiarrhythmica should be antiarrhythmics
Answer: we have amended this type error.
- Line 210/211 appears to be incomplete.
Answer: we thank you for pointing this issue out, we have completed the sentence.
Reviewer 2 Report (New Reviewer)
Comments and Suggestions for Authors
Title is informative, although authors could add that adherence to GDMT was investigated at hospital discharge and after 3 months of follow-up.
Abstract is fine. No suggestions for corrections in this section.
Guideline Directed Medical Therapy and Optimal Medical Treatment in key words seems to be pleonasm so I suggest to keep the first one.
In the Introduction please add that authors investigated patients with heart failure with LVEF under 50% (lines 84-86).
It is important that authors mentioned in the Materials and Methods part that they included patients with HFmEF and those with HFrEF (where treatment is different according to the ESC guidelines). Also, exclusion criteria were death during hospitalization and hemodynamic instability, as well as, having LVAD.
Also, it is important to name the most common adverse effects and contraindications for each of four group of medications constituting the GDMT in this cohort of patients.
I suggest explaining briefly Charlson Comorbidity Index and John Hopkins in-hospital fall risks assessment tool in part Materials and Methods (which were assessed retrospectively).
There is a missing part of the last sentence in part 3.2.1 [“For the other investigated determinants (sex, Charlson comorbidity index, polypharmacy)”]. Please add it.
In the Results part I suggest redesigning Figure 3 or split in two Figures for more precision.
In the Discussion part authors mentioned relevant studies in a few sentences.
Only 3% and 8% of patients got GDMT on discharge and after 3-months of follow up. This should be mentioned in the Conclusion part.
Study limitations and strengths are adequate.
References are up to date.
Please correct the typos.
Since authors have only 60 patients included, this is very small sample for sound conclusions, but it is a “real life study” and it points out that GDMT is poorly implemented especially in the elderly and frail patients in daily practice.
Comments on the Quality of English Languageno major concerns
Author Response
Reviewer 2:
- Title is informative, although authors could add that adherence to GDMT was investigated at hospital discharge and after 3 months of follow-up.
Answer:
We agree that this is important additional information and have added it to the title.
- Abstract is fine. No suggestions for corrections in this section.
Answer: Thank you for the positive feedback.
- Guideline Directed Medical Therapy and Optimal Medical Treatment in key words seems to be pleonasm so I suggest to keep the first one.
Answer: thank you for pointing this out. With optimal medical treatment we actually mean optimal drug dosage of heart failure medication. Therefore we have changed the keyword optimal medical treatment to Optimal drug dosage.
- In the Introduction please add that authors investigated patients with heart failure with LVEF under 50% (lines 84-86).
Answer: we agree this is important information and have added this in our introduction section.
- It is important that authors mentioned in the Materials and Methods part that they included patients with HFmEF and those with HFrEF (where treatment is different according to the ESC guidelines). Also, exclusion criteria were death during hospitalization and hemodynamic instability, as well as, having LVAD.
Answer: we agree that this is an important patient group and have therefore used these in- and exclusion criteria.
- Also, it is important to name the most common adverse effects and contraindications for each of four group of medications constituting the GDMT in this cohort of patients.
We agree this information is of added value. Therefore, we have added the most common adverse effects and contra-indications we encountered to the results section 3.4.
- I suggest explaining briefly Charlson Comorbidity Index and John Hopkins in-hospital fall risks assessment tool in part Materials and Methods (which were assessed retrospectively).
Answer: we agree that the indexes themselves and how the researchers acquired the data need further explanation. Therefore, we have added the following paragraphs to the methods section [section 2.5]
- There is a missing part of the last sentence in part 3.2.1 [“For the other investigated determinants (sex, Charlson comorbidity index, polypharmacy)”]. Please add it.
Answer: Thank you for pointing out this error, we have completed the sentence.
- In the Results part I suggest redesigning Figure 3 or split in two Figures for more precision.
Answer: we thank you for this suggestion, have split up the figure in Figure 3A (discharge) and 3B(3 month follow up) to improve the comprehensibility of the figure.
- In the Discussion part authors mentioned relevant studies in a few sentences. Only 3% and 8% of patients got GDMT on discharge and after 3-months of follow up. This should be mentioned in the Conclusion part. Study limitations and strengths are adequate.
Answer: thank you for your kind words on our discussion section. We agree that the percentage of patients who received GDMT in optimal drug dose (3 and 8%) at discharge and follow up is also important and added this to our conclusions.
- References are up to date. Please correct the typos.
Answer: We have reviewed the entire manuscript and corrected any typographical errors we encountered.
- Since authors have only 60 patients included, this is very small sample for sound conclusions, but it is a “real life study” and it points out that GDMT is poorly implemented especially in the elderly and frail patients in daily practice.
Answer: We agree that the number of included patients is very low. However, as you point out, this is a descriptive real life study with the main aim to point out the current issues encountered in daily practice. Therefore, we believe this study adds more nuance and context to current available studies on this topic
Round 2
Reviewer 1 Report (New Reviewer)
Comments and Suggestions for Authors
The authors have responded appropriately to my initial queries and have mostly amended the manuscript and supplementary material accordingly.
Figure 1 still needs to be changed to "SGLT-2 inhibitors" not just SGLT-2. Moreover, the "Ace" in the same figure needs to be all capital letters as it is an abbreviation. If there is limited space in the figure, the authors can use the abbreviations and define them in the caption below.
Author Response
Comment 1:The authors have responded appropriately to my initial queries and have mostly amended the manuscript and supplementary material accordingly.
Figure 1 still needs to be changed to "SGLT-2 inhibitors" not just SGLT-2. Moreover, the "Ace" in the same figure needs to be all capital letters as it is an abbreviation. If there is limited space in the figure, the authors can use the abbreviations and define them in the caption below.
Response 1: We appreciate the reviewer's comments. We have changed the figure according to the suggestions. We used the abbreviations to clarify SGLT2-I.
Reviewer 2 Report (New Reviewer)
Comments and Suggestions for Authors
Authors accepted all suggestions and they significantly improved the quality of the manuscript. Therefore, I suggest publishing.
Author Response
Reviewer 2: Authors accepted all suggestions and they significantly improved the quality of the manuscript. Therefore, I suggest publishing.
This manuscript is a resubmission of an earlier submission. The following is a list of the peer review reports and author responses from that submission.
Round 1
Reviewer 1 Report
Comments and Suggestions for Authors
The topic of the article is interesting and has a real clinical value. However, unfortunately, despite the many types of data processing performed by the authors, the manuscript could not be accepted in its present form. The nr. of patients is too low, and due to the huge lost to follow up, is impossible to draw pertinent conclusions regarding the subject of the paper. I would like to have again this manuscript with more patients recruted, also, with some data about the relationship of GDMT adherence and mortality/hospitalization data in this elederly population.
Reviewer 2 Report
Comments and Suggestions for Authors
Authors must be congratulated for their manuscript dealing with many hot topics, such as JF drugs and fragile patients; here my comments;
The retrospective application of the Clinical Frailty Scale is acceptable and supported by prior validation studies. However, the manuscript would benefit from a more detailed explanation of how inter-rater reliability was assessed, particularly regarding the consistency between the two evaluators.
In addition, the rationale behind the selection of variables for logistic regression analysis is insufficiently explained. Although the authors mention adjustments for age and sex “when power allowed,” the manuscript does not clearly specify which analyses incorporated logistic regression or the criteria used for model inclusion.
It is recommended that future research efforts include systematic documentation of the reasons for non-adherence to GDMT in clinical records. Alternatively, physician interviews could be conducted to supplement chart reviews and capture undocumented rationale.
In the discussion are strongly encouraged to include the latest evidences of pleiotropic effects of SGLT2i drugs (doi: 10.1002/ehf2.15223) that could be particular useful in the setting of fragile patients.
Comments on the Quality of English LanguageLastly, there are some grammatical inconsistencies throughout the manuscript:
“patients was lost to follow-up” should be corrected to “patients were lost to follow-up.”
“out of fear for adverse effects” could be more accurately phrased as “due to concerns about adverse effects.”
Moreover the discussion section would benefit from improved consistency in verb tense usage.
Reviewer 3 Report
Comments and Suggestions for Authors
Interesting article on a well-known topic interesting article on a well-known topic
Small sample size, high loss to follow-up, retrospective design, incomplete data, heterogeneity (HFrEF/HFmrEF), lack of dynamic clinical variables, no multivariable analysis, and absence of clinical outcomes.